# GENERATIVE MODEL BASED ON MINIMIZING EXACT EMPIRICAL WASSERSTEIN DISTANCE

## ABSTRACT

Generative Adversarial Networks (GANs) are a very powerful framework for generative modeling. However, they are often hard to train, and learning of GANs often becomes unstable. Wasserstein GAN (WGAN) is a promising framework to deal with the instability problem as it has a good convergence property. One drawback of the WGAN is that it evaluates the Wasserstein distance in the dual domain, which requires some approximation, so that it may fail to optimize the true Wasserstein distance. In this paper, we propose evaluating the exact empirical optimal transport cost efficiently in the primal domain and performing gradient descent with respect to its derivative to train the generator network. Experiments on the MNIST dataset show that our method is significantly stable to converge, and achieves the lowest Wasserstein distance among the WGAN variants at the cost of some sharpness of generated images. Experiments on the 8-Gaussian toy dataset show that better gradients for the generator are obtained in our method. In addition, the proposed method enables more flexible generative modeling than WGAN.

## 1 INTRODUCTION

Generative Adversarial Networks (GANs) (Goodfellow et al., 2014) are a powerful framework of generative modeling which is formulated as a minimax game between two networks: A generator network generates fake-data from some noise source and a discriminator network discriminates between fake-data and real-data. GANs can generate much more realistic images than other generative models like variational autoencoder (Kingma & Welling, 2014) or autoregressive models (van den Oord et al., 2016), and have been widely used in high-resolution image generation (Karras et al., 2018), image inpainting (Yu et al., 2018), image-to-image translation (Isola et al., 2017), to mention a few. However, GANs are often hard to train, and various ways to stabilize training have been proposed by many recent works. Nonetheless, consistently stable training of GANs remains an open problem.

GANs employ the Jensen-Shannon (JS) divergence to measure the distance between the distributions of real-data and fake-data (Goodfellow et al., 2014). Arjovsky et al. (2017) provided an analysis of various distances and divergence measures between two probability distributions in view of their use as loss functions of GANs, and proposed Wasserstein GAN (WGAN) which has better theoretical properties than the original GANs. WGAN requires that the discriminator (called the *critic* in Arjovsky et al. (2017)) must lie within the space of 1-Lipschitz functions to evaluate the Wasserstein distance via the Kantorovich-Rubinstein dual formulation. Arjovsky et al. (2017) further proposed implementing the critic with a deep neural network and applied weight clipping in order to ensure that the critic satisfies the Lipschitz condition. However, weight clipping limits the critic's function space and can cause gradients in the critic to explode or vanish if the clipping parameters are not carefully chosen (Arjovsky et al., 2017; Gulrajani et al., 2017). WGAN-GP (Gulrajani et al., 2017) and Spectral Normalization (SN) (Miyato et al., 2018) apply regularization and normalization, respectively, on the critic trying to make the critic 1-Lipschitz, but they fail to optimize the true Wasserstein distance.

In the latest work, Liu et al. (2018) proposed a new WGAN variant to evaluate the exact empirical Wasserstein distance. They evaluate the empirical Wasserstein distance between the empirical distributions of real-data and fake-data in the discrete case of the Kantorovich-Rubinstein dual for-

mulation, which can be solved efficiently because the dual problem becomes a finite-dimensional linear-programming problem. The generator network is trained using the critic network learnt to approximate the solution of the dual problem. However, the problem of approximation error by the critic network remains. In this paper, we propose a new generative model without the critic, which learns by directly evaluating gradient of the exact empirical optimal transport cost in the primal domain. The proposed method corresponds to stochastic gradient descent of the optimal transport cost.

## 2 WASSERSTEIN GAN

Arjovsky et al. (2017) argued that JS divergences are potentially not continuous with respect to the generator's parameters, leading to GANs training difficulty. They proposed instead using the Wasserstein-1 distance $W_1(q, p)$, which is defined as the minimum cost of transporting mass in order to transform the distribution $q$ into the distribution $p$. Under mild assumptions, $W_1(q, p)$ is continuous everywhere and differentiable almost everywhere.

The WGAN objective function is constructed using the Kantorovich-Rubinstein duality (Villani, 2009, Chapter 5) as

$$W_1(\mathbb{P}_r, \mathbb{P}_g) = \max_{D \in \mathcal{D}} \left\{ \mathbb{E}_{\boldsymbol{x} \sim \mathbb{P}_r} [D(\boldsymbol{x})] - \mathbb{E}_{\boldsymbol{y} \sim \mathbb{P}_g} [D(\boldsymbol{y})] \right\}, \tag{1}$$

to obtain

$$\min_{G} \max_{D \in \mathcal{D}} \left\{ \mathbb{E}_{\boldsymbol{x} \sim \mathbb{P}_r} [D(\boldsymbol{x})] - \mathbb{E}_{\boldsymbol{y} \sim \mathbb{P}_g} [D(\boldsymbol{y})] \right\}, \tag{2}$$

where $\mathcal{D}$ is the set of all 1-Lipschitz functions, where $\mathbb{P}_r$ is the real-data distribution, and where $\mathbb{P}_g$ is the generator distribution implicitly defined by $\boldsymbol{y} = G(\boldsymbol{z}), \boldsymbol{z} \sim p(\boldsymbol{z})$. Minimization of this objective function with respect to $G$ with optimal $D$ is equivalent to minimizing $W_1(\mathbb{P}_r, \mathbb{P}_g)$.

Arjovsky et al. (2017) further proposed implementing the critic $D$ in terms of a deep neural network with weight clipping. Weight clipping keeps the weight parameter of the network lying in a compact space, thereby ensuring the desired Lipschitz condition. For a fixed network architecture, however, weight clipping may significantly limit the function space to a quite small fraction of all possible 1-Lipschitz functions representable by networks with the prescribed architecture.

## 3 RELATED WORKS

Gulrajani et al. (2017) proposed introduction of gradient penalty (GP) to the WGAN objective function in place of the 1-Lipschitz condition in the Kantorovich-Rubinstein dual formulation, in order to explicitly encourage the critic to have gradients with magnitude equal to 1. Since enforcing the constraint of unit-norm gradient everywhere is intractable, they proposed enforcing the constraint only along straight line segments, each connecting a real-data point and a fake-data point. The resulting learning scheme, which is called the WGAN-GP, was shown to perform well experimentally. It was pointed out, however (Miyato et al., 2018), that WGAN-GP is susceptible to destabilization due to gradual changes of the support of the generator distribution as learning progresses. Furthermore, the critic can easily violate the Lipschitz condition in practice, so that there is no guarantee that WGAN-GP optimizes the true Wasserstein distance.

SN, proposed by Miyato et al. (2018), is based on the observation that the Lipschitz norm of a critic represented by a multilayer neural network is bounded from above by the product, across all layers, of the Lipschitz norms of the activation functions and the spectral norms of the weight matrices, and normalizes each of the weight matrices with its spectral norm to ensure the resulting critic to satisfy the desired Lipschitz condition. It is well known that, for any $m \times n$ matrix $W = (w_{ij})$, the max norm $\|W\|_{\max} = \max\{|w_{ij}|\}$ and the spectral norm $\sigma(W)$ satisfy the inequality $\|W\|_{\max} \leq \sigma(W) \leq \sqrt{mn}\|W\|_{\max}$. This implies that the bound of the Lipschitz constant provided via weight clipping can be loose compared with that via SN. In other words, SN is expected to provide a much tighter bound for the Lipschitz condition than weight clipping, and accordingly, the function space for the critic under SN is larger than that under weight clipping. The function space under SN is, however, still a subset of the set of all functions satisfying the Lipschitz condition, and consequently, the resulting estimate for the Wasserstein distance is a lower bound of the true Wasserstein distance. Furthermore, one cannot tell within the framework of SN how good the estimate is.

Liu et al. (2018) proposed a new formulation to evaluate the Wasserstein distance, which is equivalent to the discrete case of the Kantorovich-Rubinstein dual formulation under a mild assumption and is more tractable due to obviating the need for the Lipschitz condition. This problem is solved in a two-step fashion, and thus the method proposed in Liu et al. (2018) is called the WGAN-TS. First, one estimates the Wasserstein distance on the basis of finite real- and fake-data points. The empirical Wasserstein distance is evaluated exactly via solving the linear-programming version of the Kantorovich-Rubinstein dual to obtain the optimizer. Second, one approximates the optimizer obtained in the first step via regression using a deep neural network to parameterize the critic and obtains its gradient. WGAN-TS can evaluate the Wasserstein distance more accurately than WGAN, WGAN-GP and WGAN-SN (WGAN with SN), but there are not only approximation errors from using finite samples to evaluate the empirical Wasserstein distance but also those from deep regression in the second step, resulting in not being able to minimize the Wasserstein distance directly.

## 4 PROPOSED METHOD

The proposed method in this paper is based on the fact that the optimal transport cost between two probability distributions can be evaluated efficiently when the distributions are uniform over finite sets of the same cardinality. Our proposal is to evaluate empirical optimal transport costs on the basis of equal-size sample datasets of real- and fake-data points.

The optimal transport cost between the real-data distribution $\mathbb{P}_r$ and the generator distribution $\mathbb{P}_g$ is defined as

$$C(\mathbb{P}_r, \mathbb{P}_g) = \inf_{\gamma \in \Pi(\mathbb{P}_r, \mathbb{P}_g)} \mathbb{E}_{(x,y)\sim\gamma}[c(x,y)], \tag{3}$$

where $c(x,y)$ is the cost of transporting one unit mass from $x$ to $y$, assumed differentiable with respect to its arguments almost everywhere, and where $\Pi(\mathbb{P}_r, \mathbb{P}_g)$ denotes the set of all couplings between $\mathbb{P}_r$ and $\mathbb{P}_g$, that is, all joint probability distributions that have marginals $\mathbb{P}_r$ and $\mathbb{P}_g$.

Let $D = \{x_j | x_j \sim \mathbb{P}_r(x)\}$ be a dataset consisting of independent and identically-distributed (iid) real-data points, and $F = \{y_i | y_i \sim \mathbb{P}_g(y)\}$ be a dataset consisting of iid fake-data points sampled from the generator. Let $\mathbb{P}_D$ and $\mathbb{P}_F$ be the empirical distributions defined by the datasets $D$ and $F$, respectively. We further assume in the following that $|D| = |F| = N$ holds. The empirical optimal transport cost $\hat{C}(D, F) = C(\mathbb{P}_D, \mathbb{P}_F)$ between the two datasets $D$ and $F$ is formulated as a linear-programming problem, as

$$\hat{C}(D, F) = C(\mathbb{P}_D, \mathbb{P}_F) = \frac{1}{N} \min_M \sum_{i=1}^{N} \sum_{j=1}^{N} M_{i,j} c(x_j, y_i) \tag{4}$$

$$\text{s.t. } \sum_{j=1}^{N} M_{i,j} = 1, \forall i \in \{1, \ldots, N\}, \tag{5}$$

$$\sum_{i=1}^{N} M_{i,j} = 1, \forall j \in \{1, \ldots, N\}, \tag{6}$$

$$M_{i,j} \geq 0, \forall i \in \{1, \ldots, N\}, \forall j \in \{1, \ldots, N\}. \tag{7}$$

It is known (Villani, 2003) that the linear-programming problem (4)–(7) admits solutions which are permutation matrices. One can then replace the constraints $M_{i,j} \geq 0$ in (7) with $M_{i,j} \in \{0, 1\}$ without affecting the optimality. The resulting optimization problem is what is called the linear sum assignment problem, which can be solved more efficiently than the original linear-programming problem. As far as the authors' knowledge, the most efficient algorithm to date for solving a linear sum assignment problem has time complexity of $\mathcal{O}(N^{2.5} \log(N\mathcal{C}))$, where $\mathcal{C} = \max_{i,j} c(x_j, y_i)$ when one scales up the costs $\{c(x_j, y_i) | x_j \in D, y_i \in F\}$ to integers (Burkard et al., 2012, Chapter 4).

This is a problem to find the optimal transport plan, where $M_{i,j} = 1$ is corresponding to transporting fake-data point $y_i \in F$ to real-data point $x_j \in D$, and where the objective is to minimize the average transport cost $N^{-1} \sum_{i=1}^{N} \sum_{j=1}^{N} M_{i,j} c(x_j, y_i)$. Figure 1 shows a two-dimensional example of this problem and its solution.

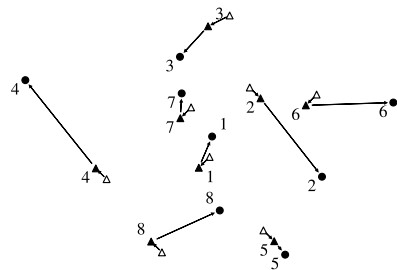

Figure 1: Two-dimensional example of optimal transport problem (4)–(7) with $N = 8$. Circles $\bullet$ represent real-data points in $D$ and triangles $\blacktriangle$ represent fake-data points in $F$. Arrows between circles $\bullet$ and filled triangles $\blacktriangle$ show the optimal transport plan $M^*$ with $c(x, y) = \|x - y\|_2$, which is an identity matrix. Arrows between open $\triangle$ and filled $\blacktriangle$ triangles show small perturbations of $F$, which do not change $M^*$.

One requires evaluations not only of the optimal transport cost $C(\mathbb{P}_r, \mathbb{P}_g)$ but also of its derivative in order to perform learning of the generator with backpropagation. Let $\theta$ denote the parameter of the generator, and let $\partial_\theta C$ denote the derivative of the optimal transport cost $C$ with respect to $\theta$. Conditional on $\boldsymbol{z}$, the generator output $G(\boldsymbol{z})$ is a function of $\theta$. Hence, in order to estimate $\partial_\theta C$, in our framework one has to evaluate $\partial_\theta \hat{C}$. In general, it is difficult to differentiate (4) with respect to generator output $y_i$, as the optimal transport plan $M^*$ can be highly dependent on $y_i$. Under the assumption $|D| = |F| = N$ which we adopt here, however, the feasible set for $M$ is the set of all permutation matrices and is a finite set. It then follows that, as a generic property, the optimal transport plan $M^*$ is unchanged under small enough perturbations of $F$ (see Figure 1). We take advantage of this fact and regard $M^*$ as independent of $y_i$. Now that differentiation of (4) becomes tractable, we use (4) as the loss function of the generator and update the generator with the direct gradient of the empirical optimal transport cost, as $\partial_\theta \hat{C} = N^{-1} \sum_{i,j=1}^{N} M_{i,j}^* \partial_{y_i} c(x_j, y_i) \partial_\theta G(z_i)$.

Although the framework described so far is applicable to any optimal transport cost, several desirable properties can be stated if one specializes in the Wasserstein distance. Assume, for a given $p \geq 1$, that the real-data distribution $\mathbb{P}_r$ and the generator distribution $\mathbb{P}_g$ have finite moments of order $p$. The Wasserstein-$p$ distance between $\mathbb{P}_r$ and $\mathbb{P}_g$ is defined in terms of the optimal transport cost with $c(x, y) = \|x - y\|^p$ as

$$W_p(\mathbb{P}_r, \mathbb{P}_g) = C(\mathbb{P}_r, \mathbb{P}_g)^{1/p}. \tag{8}$$

Due to the law of large numbers, the empirical distributions $\mathbb{P}_D$ and $\mathbb{P}_F$ converge weakly to $\mathbb{P}_r$ and $\mathbb{P}_g$, respectively, as $N \to \infty$. It is also known (Villani, 2009, Theorem 6.9) that the Wasserstein-$p$ distance $W_p$ metrizes the space of probability measures with finite moments of order $p$. Consequently, the empirical Wasserstein distance $\hat{W}_p(D, F)$ is a consistent estimator of the true Wasserstein distance $W_p(\mathbb{P}_r, \mathbb{P}_g)$. Furthermore, with the upper bound of the error of the estimator

$$|\hat{W}_p(D, F) - W_p(\mathbb{P}_r, \mathbb{P}_g)| \leq W_p(\mathbb{P}_D, \mathbb{P}_r) + W_p(\mathbb{P}_F, \mathbb{P}_g), \tag{9}$$

which is derived on the basis of the triangle inequality, as well as with the upper bounds available for expectations of $W_p(\mathbb{P}_D, \mathbb{P}_r)$ and $W_p(\mathbb{P}_F, \mathbb{P}_g)$ under mild conditions (Weed & Bach, 2017), one can see that $\hat{W}_p(D, F)$ is an asymptotically unbiased estimator of $W_p(\mathbb{P}_r, \mathbb{P}_g)$.

Note that our method can directly evaluate the empirical Wasserstein distance without recourse to the Kantorovich-Rubinstein dual. Hence, our method does not use a critic and is therefore no longer a GAN. It is also applicable to any optimal transport cost. We summarize the proposed method in Algorithm 1.

## 5 EXPERIMENTS

### 5.1 RESULTS ON MNIST WITH CONVOLUTIONAL NEURAL NETWORK

We first show experimental results on the MNIST dataset of handwritten digits. In this experiment, we resized the images to resolution $64 \times 64$ so that we can use the convolutional neural networks

---

**Algorithm 1** The proposed method.

---

1: Input: Real-data samples $X_{\text{real}}$, batch size $N$, Adam parameters $\alpha, \beta_1, \beta_2$
2: Output: $G_\theta$
3: Initialize $\theta$.
4: **while** $\theta$ has not converged **do**
5:     Sample $\{x_i\}_{i \in \{1,\dots,N\}} \sim X_{\text{real}}$ from real-data.
6:     Sample $\{z_j\}_{j \in \{1,\dots,N\}} \sim p(z)$ from random noises.
7:     Let $y_j = G_\theta(z_j)$, $\forall j \in \{1,\dots,N\}$.
8:     Solve (4)–7 to obtain $M^*$.
9:     $g_\theta \leftarrow \partial_\theta \hat{C} = N^{-1} \sum_{i,j=1}^N M_{i,j}^* \partial_{y_i} c(x_j, y_i) \partial_\theta G_\theta(z_i)$
10:     $\theta \leftarrow \text{Adam}(g_\theta, \theta, \alpha, \beta_1, \beta_2)$
11: **end while**

---

Table 1: Comparison on MNIST. Empirical Wasserstein distance (EWD), and computation time per generator update for each method. Each metric represents the average and standard deviation over 3 out of 5 trials, excluding the maximum and minimum. Lower is better for EWD. WGAN-TS* represents WGAN-TS without weight scaling. In WGAN-GP, WGAN-SN and WGAN-TS*, training can suddenly deteriorate. Thus, we used early stopping based on EWD.

| Method | EWD | Time [ms/iter] |
|---|---|---|
| WGAN | $744.8 \pm 5.2$ | $1547 \pm 18.2$ |
| WGAN-GP | $811.9 \pm 31.7$ | $2867 \pm 43.6$ |
| WGAN-SN | $888.5 \pm 30.1$ | $1839 \pm 44.9$ |
| WGAN-TS* | $714.5 \pm 17.7$ | $886.0 \pm 10.0$ |
| WGAN-TS | $761.2 \pm 4.5$ | $896.8 \pm 9.44$ |
| Proposed | $\mathbf{600.5 \pm 10.8}$ | $\mathbf{148.4 \pm 3.58}$ |

described in Appendix A.1 as the critic and the generator. In all methods, the batch size was set to 64 and the prior noise distribution was the 100-dimensional standard normal distribution. The maximum number of iterations in training of the generator was set to 30,000. The Wasserstein-1 distance with $c(x, y) = \|x - y\|_1$ was used. More detailed settings are described in Appendix B.1.

Although several performance metrics have been proposed and are commonly used to evaluate variants of WGAN, we have decided to use the empirical Wasserstein distance (EWD) to compare performance of all methods. It is because all the methods adopt objective functions that are based on the Wasserstein distance, and because EWD is a consistent and asymptotically unbiased estimator of the Wasserstein distance and can efficiently be evaluated, as discussed in Section 4. Table 1 shows EWD evaluated with 256 samples and computation time per generator update for each method. For reference, performance comparison with the Fréchet Inception Distance (Heusel et al., 2017) and the Inception Score (Salimans et al., 2016), which are commonly used as performance measures to evaluate GANs using feature space embedding with an inception model, is shown in Appendix C. The proposed method achieved a remarkably small EWD and computational cost compared with the variants of WGAN. Our method required the lowest computational cost in this experimental setting mainly because it does not use the critic. Although we think that the batch size used in the experiment of the proposed method was appropriate since the proposed method achieved lower EWD, if a larger batch size would be required in training, it will take much longer time to solve the linear sum assignment problem (4)–(7).

We further investigated behaviors of the methods compared in more detail, on the basis of EWD. WGAN-SN failed to learn. The loss function of the critic showed divergent movement toward $-\infty$, and the behaviors of EWD in different trials were different even though the behaviors of the critic loss were the same (Figure 2 (a) and (b)). WGAN training never failed in 5 trials, and EWD improved stably without sudden deterioration. Although training with WGAN-GP proceeded favorably in initial stages, at certain points the gradient penalty term started to increase, causing EWD to deteriorate (Figure 2 (c)). This happened in all 5 trials. Since gradient penalty is a weaker restriction than weight clipping, the critic may be more likely to cause extreme behaviors. We examined both WGAN-TS with and without weight scaling. Whereas WGAN-TS with weight scaling did not fail

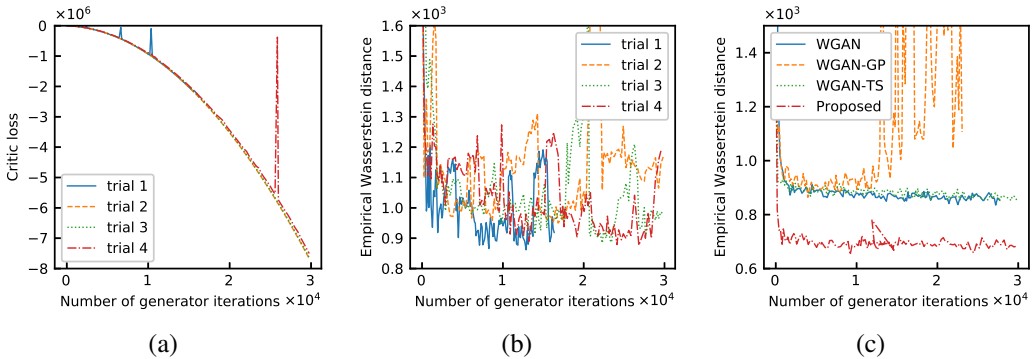

Figure 2: Training behaviors of the methods experimented. (a) Critic loss versus the number of generator iterations, and (b) EWD versus the number of generator iterations in four trials of WGAN-SN. (c) EWD versus the number of generator iterations, averaged over 5 trials, in WGAN, WGAN-GP, WGAN-TS, and the proposed method.

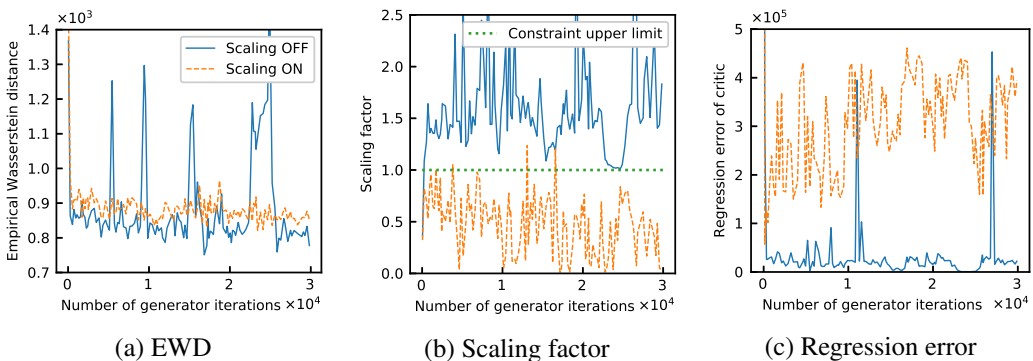

Figure 3: Typical behavior of WGAN-TS in training on the MNIST dataset with and without weight scaling. (a) EWD. (b) Scaling factor, defined as $\beta = \sup\{1, \sup\{\frac{D(y_i)-D(G(z_j))}{||y_i-G(z_j)||}, ||y_i - G(z_j)|| > 0\}\}$ in Liu et al. (2018, Section 4.1). If $\beta \leq 0$, the critic satisfies the 1-Lipschitz constraint. (c) Regression error of the critic.

in training but achieved higher EWD than WGAN, WGAN-TS without weight scaling achieved lower EWD than WGAN at the cost of the stability of training (Figure 3). The proposed method was trained stably and never failed in 5 trials.

As mentioned in Section 3, the critic in WGAN-TS simply regresses the optimizer of the empirical version of the Kantorovich-Rubinstein dual. Thus, there is no guarantee that the critic will satisfy the 1-Lipschitz condition. Liu et al. (2018) pointed out that it is indeed practically problematic with WGAN-TS, and proposed weight scaling to ensure that the critic satisfies the desired condition. We have empirically found, however, that weight scaling exhibited the following trade-off (Figure 3). Without weight scaling, training of WGAN-TS suddenly deteriorated in some trials because the critic came to not satisfy the Lipschitz condition. With weight scaling, on the other hand, the regression error of the critic with respect to the solution increased and the EWD became worse. The proposed method directly solves the empirical version of the optimal transport problem in the primal domain, so that it is free from such trade-off.

Figure 4 shows fake-data images generated by the generators trained with WGAN, WGAN-GP, WGAN-TS, and the proposed method. Although one can identify the digits for the generated images with the proposed method most easily, these images are less sharp. Among the generated images with the other methods, one can notice several images which have almost the same appearance as real-data images, whereas in the proposed method, such fake-data images are not seen and images that seem averaged real-data images belonging to the same class often appear. This might imply that

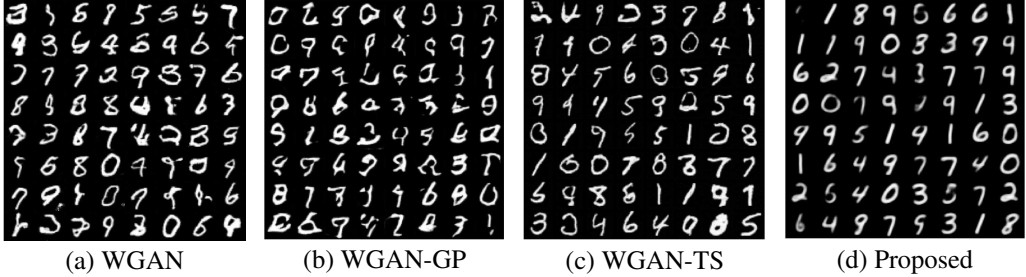

(a) WGAN      (b) WGAN-GP      (c) WGAN-TS      (d) Proposed

Figure 4: Examples of fake-data images generated by the generators trained on the MNIST dataset with the four methods compared.

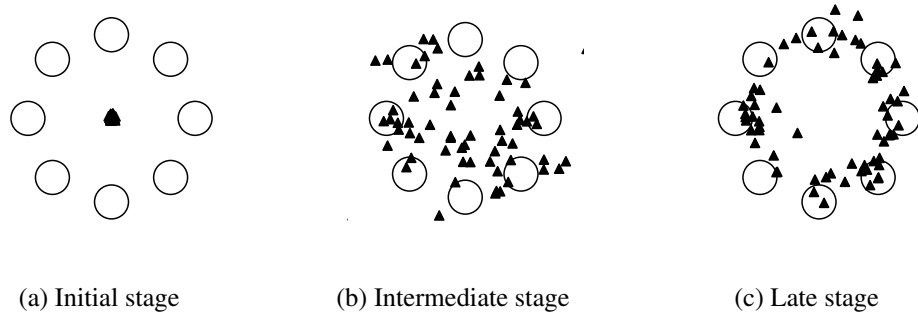

(a) Initial stage      (b) Intermediate stage      (c) Late stage

Figure 5: Generator distribution in successful trial with the proposed method on the 8-Gaussian toy dataset. Eight large circles ◯ represent positions of Gaussian components of real-data distribution, and filled triangles ▲ are fake-data points. Snapshots in (a) initial stage, (b) intermediate stage, and (c) late stage of training are shown.

merely minimizing the Wasserstein distance between the real-data distribution and the generator distribution in the raw-image space may not necessarily produce realistic images.

## 5.2 GRADIENT OPTIMALITY OF GENERATOR

We next observed how the generator distribution is updated in order to compare the proposed method with variants of WGAN in terms of the gradients provided. Figure 5 shows typical behavior of the generator distribution trained with the proposed method on the 8-Gaussian toy dataset. The 8-Gaussian toy dataset and experimental settings are described in Appendix B.2. One can observe that, as training progresses, the generator distribution comes closer to the real-data distribution. Figure 6 shows comparison of the behaviors of the proposed method, WGAN-GP, and WGAN-TS. We excluded WGAN and WGAN-SN from this comparison: WGAN tended to yield generator distributions that concentrated around a single Gaussian component, and hence training did not progress well. WGAN-SN could not correctly evaluate the Wasserstein distance as in the experiment on the MNIST dataset.

One can observe in Figure 6 that directions of sample updates are diverse in the proposed method, especially in later stages of training, and that the sample update directions tend to be aligned with the optimal gradient directions. These behaviors will be helpful for the generator to learn the real-data distribution efficiently. In WGAN-GP and WGAN-TS, on the other hand, the sample update directions exhibit less diversity and less alignment with the optimal gradient directions, which would make the generator distribution difficult to spread and would slow training. One would be able to ascribe such behaviors to poor quality of the critic: Those behaviors would arise when the generator learns on the basis of unreliable gradient information provided by the critic without learning sufficiently to accurately evaluate the Wasserstein distance. If one would increase the number $n_c$ of critic iterations per generator iteration in order to train the critic better, the total computational cost of training would increase. In fact, $n_c = 5$ is recommended in practice and has commonly been used

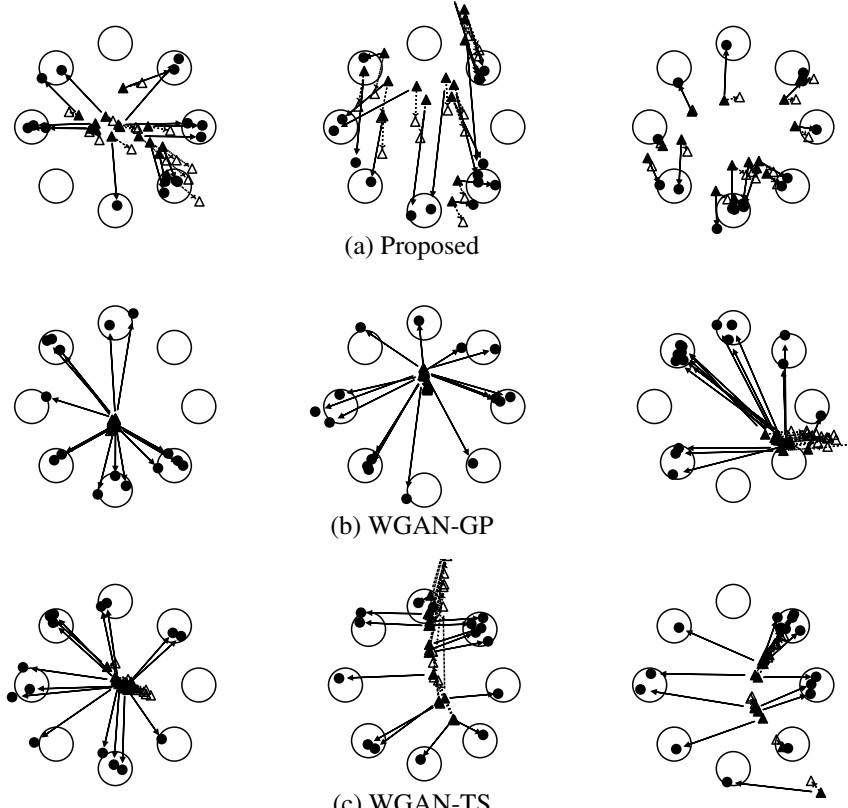

(a) Proposed

(b) WGAN-GP

(c) WGAN-TS

Figure 6: Behaviors of updating in training on the 8-Gaussian toy dataset with (a) the proposed method, (b) WGAN-GP, and (c) WGAN-TS. Eight large circles ◯ represent positions of Gaussian components of real-data distribution. Filled circles ● are real-data points, filled triangles ▲ are pre-updated fake-data points, and open triangles △ are post-updated fake-data points. Solid and dotted arrows indicate optimal gradient directions of fake-data points and directions of their actual updates, respectively. Left, center, and right panels are at iterations 10, 20, and 30, respectively.

in WGAN (Arjovsky et al., 2017) and its variants because the improvement in learning of the critic is thought to be small relative to increase in computational cost. In reality, however, 5 iterations would not be sufficient for the critic to learn, and this might be a principal reason for the critic to provide poor gradient information to the generator in the variants of WGAN.

## 6 CONCLUSION

We have proposed a new generative model that learns by directly minimizing exact empirical Wasserstein distance between the real-data distribution and the generator distribution. Since the proposed method does not suffer from the constraints on the transport cost and the 1-Lipschitz constraint imposed on WGAN by solving the optimal transport problem in the primal domain instead of the dual domain, one can construct more flexible generative modeling. The proposed method provides the generator with better gradient information to minimize the Wasserstein distance (Section 5.2) and achieved smaller empirical Wasserstein distance with lower computational cost (Section 5.1) than any other compared variants of WGAN. In the future work, we would like to investigate the behavior of the proposed method when transport cost is defined in the feature space embedded by an appropriate inception model.

ACKNOWLEDGMENTS

Support of anonymous funding agencies is acknowledged.

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

# Appendix

## A NETWORK ARCHITECTURES

### A.1 CONVOLUTIONAL NEURAL NETWORKS

We show in Table 2 the network architecture used in the experiment on the MNIST dataset in Section 5.1. The generator network receives a 100-dimensional noise vector generated from the standard normal distribution as an input. The noise vector is passed through the fully-connected layer and reshaped to $4 \times 4$ feature maps. Then they are passed through four transposed convolution layers with $5 \times 5$ kernels, stride 2 and no biases (since performance was empirically almost the same with or without biases, we took the simpler option of not considering biases), where the resolution of feature maps is doubled and the number of them is halved except for the last layer.

The critic network is basically the reverse of the generator network. A convolution layer is used instead of a transposed convolution layer in the critic. After the last convolution layer, the feature maps are flattened into a vector and passed through the fully-connected layer.

We employed batch normalization (Ioffe & Szegedy, 2015) in all intermediate layers in both of the generator and the critic. Rectified linear unit (ReLU) was used as the activation function in all but the last layers. As the activation function in the last layer, the hyperbolic tangent function and the identity function were used for the generator and for the critic, respectively.

### A.2 FULLY-CONNECTED NEURAL NETWORKS

We show in Table 3 the network architecture used in the experiment on the 8-Gaussian toy dataset in Section 5.2. The generator network architecture receives a 100-dimensional noise vector as in the experiment on the MNIST dataset. The noise vector is passed through the four fully-connected layers with biases and mapped to a two-dimensional space. The critic network is likewise the reverse of the generator network.

## B DETAILED EXPERIMENTAL SETTING

### B.1 MNIST DATASET

The MNIST dataset of handwritten digits used in the experiment in Section 5.1 contains 60,000 two-dimensional images of handwritten digits with resolution $28 \times 28$.

We used default parameter settings decided by the proposers of the respective methods. We used RMSProp (Hinton et al., 2012) with learning rate $5e-5$ for the critic and the generator in WGAN. The weight clipping parameter $c$ was set to 0.01. We used Adam (Kingma & Ba, 2015) with learning rate $1e-4, \beta_1 = 0.5, \beta_2 = 0.999$ in the other methods. $\lambda_{\mathrm{gp}}$ in WGAN-GP was set to 10. In the methods with the critic, the number $n_c$ of critic iterations per generator iteration was set to 5.

### B.2 8-GAUUSIAN TOY DATASET

The 8-Gaussian toy dataset used in the experiment in Section 5.2 is a two-dimensional synthetic dataset, which contains real-data sampled from the Gaussian mixture distribution with 8 centers equally distant from the origin and unit variance as the real-data distribution. The centers of the 8 Gaussian component distributions are $(\pm 10, 0)$, $(0, \pm 10)$, and $(\pm 10/\sqrt{2}, \pm 10/\sqrt{2})$. $30,000$ samples were generated in advance before training and were used as the real-data samples.

In all methods, the batch size was set to $64$ and the maximum number of iterations in training the generator was set to 1,000. WGAN and WGAN-SN could not learn well with this dataset, even though we considered several parameter sets. We used Adam with learning rate $1e-3, \beta_1 = 0.5, \beta_2 = 0.999$ for WGAN-GP, WGAN-TS and the proposed method. $\lambda_{\mathrm{gp}}$ in WGAN-GP was set to 10. In the methods with the critic, the number $n_c$ of critic iterations was set to 5.

Table 2: Convolutional Neural Network

| Generator | Output shape | Parameters | Activation |
|---|---|---|---|
| Noise vector | (100) | - | - |
| Fully-connected | (8096) | 809.6k | ReLU |
| Reshape | (4, 4, 1024) | - | - |
| BatchNorm. | - | - | - |
| TransposedConv. $5 \times 5$ | (8, 8, 512) | 13.1M | ReLU |
| BatchNorm. | - | - | - |
| TransposedConv. $5 \times 5$ | (16, 16, 256) | 3.3M | ReLU |
| BatchNorm. | - | - | - |
| TransposedConv. $5 \times 5$ | (32, 32, 128) | 819.2k | ReLU |
| BatchNorm. | - | - | - |
| TransposedConv. $5 \times 5$ | (64, 64, 1) | 3.2k | tanh |
| Total trainable parameters | - | **18.0M** | - |
| **Critic** | Output shape | Parameters | Activation |
| Input image | (64, 64, 1) | - | - |
| Conv. $5 \times 5$ | (32, 32, 128) | 3.2k | ReLU |
| BatchNorm. | - | - | - |
| Conv. $5 \times 5$ | (16, 16, 256) | 819.2k | ReLU |
| BatchNorm. | - | - | - |
| Conv. $5 \times 5$ | (8, 8, 512) | 3.3M | ReLU |
| BatchNorm. | - | - | - |
| Conv. $5 \times 5$ | (4, 4, 1024) | 13.1M | ReLU |
| BatchNorm. | - | - | - |
| Flatten | (8096) | - | - |
| Fully-connected | (1) | 8.1k | - |
| Total trainable parameters | - | **17.2M** | - |

Table 3: Fully-connected Neural Network

| Generator | Output shape | Parameters | Activation |
|---|---|---|---|
| Noise vector | (100) | - | - |
| Fully-connected | (512) | 51.7k | ReLU |
| Fully-connected | (512) | 262.7k | ReLU |
| Fully-connected | (512) | 262.7k | ReLU |
| Fully-connected | (2)) | 1.0k | - |
| Total trainable parameters | - | **5.781M** | - |
| **Critic** | Output shape | Parameters | Activation |
| Input points | (2) | - | - |
| Fully-connected | (512) | 1.54k | ReLU |
| Fully-connected | (512) | 262.7k | ReLU |
| Fully-connected | (512) | 262.7k | ReLU |
| Fully-connected | (1) | 1.02k | - |
| Total trainable parameters | - | **5.371M** | - |

Table 4: Comparison on the MNIST dataset. Fréchet Inception Distance (FID), Inception Score (IS) Each metric represents the average and standard deviation over 3 out of 5 trials, excluding the maximum and minimum. Lower is better for FID. Higher is better for IS. FID and IS were calculated with $50,000$ samples. WGAN-TS* represents WGAN-TS without weight scaling.

| Method | FID | IS |
|---|---|---|
| WGAN | $\mathbf{0.41 \pm 0.02}$ | $2.15 \pm 0.02$ |
| WGAN-GP | $3.97 \pm 2.49$ | $2.15 \pm 0.09$ |
| WGAN-SN | $4.01 \pm 1.36$ | $2.16 \pm 0.19$ |
| WGAN-TS* | $0.74 \pm 0.23$ | $2.07 \pm 0.05$ |
| WGAN-TS | $0.68 \pm 0.06$ | $2.16 \pm 0.01$ |
| Proposed | $5.57 \pm 0.14$ | $\mathbf{2.33 \pm 0.02}$ |

### B.3 EXECUTION ENVIRONMENT

All the numerical experiments in this paper were executed on a computer with an Intel Core i7-6850K CPU (3.60 GHz, 6 cores) and 32 GB RAM, and with four GeForce GTX 1080 graphics cards installed. Linear sum assignment problems were solved using the Hungarian algorithm, which has time complexity of $\mathcal{O}(N^3)$. Codes used in the experiments were written in tensorflow 1.10.1 on python 3.6.0, with eager execution enabled.

## C EVALUATION WITH FID AND IS ON MNIST DATASET

We show the result of evaluation of the experimented methods with FID and IS in Table 4. Both FID and IS are commonly used to evaluate GANs.

FID calculates the distance between the set of real-data points and that of fake-data points. The smaller the distance is, the better the fake-data points are judged. Assuming that the vector obtained from a fake- or real-data point through the inception model follows a multivariate Gaussian distribution, FID is defined by the following equation:

$$\text{FID}^2 = \|\mu_1 - \mu_2\|_2^2 + \text{tr}\left(\Sigma_1 + \Sigma_2 - 2(\Sigma_1\Sigma_2)^{\frac{1}{2}}\right), \tag{10}$$

where $(\mu_i, \Sigma_i)$ is the mean vector and the covariance matrix for dataset $i$, evaluated in the feature space embedded with inception scores. It is nothing but the square of the Wasserstein-2 distance between two multivariate Gaussian distributions with parameters $(\mu_1, \Sigma_1)$ and $(\mu_2, \Sigma_2)$, respectively.

IS is a metric to evaluate only the set of fake-data points. Let $x_i$ be a data point, $y$ be the label of $x_i$ in the data identification task for which the inception model was trained, $p(y|x_i)$ be the probability of label $y$ obtained by inputting $x_i$ to the inception model. Letting $X$ be the set of all data points used for calculating the score, the marginal probability of label $y$ is $p(y) = \frac{1}{|X|}\sum_{x_i \in X} p(y|x_i)$. IS is defined by the following equation:

$$\text{IS} = \exp\left(\frac{1}{|X|}\sum_{x_i \in X} \text{KL}\left(p(y|x_i)\|p(y)\right)\right), \tag{11}$$

where KL is Kullback–Leibler divergence. IS is designed to be high as the data points are easy to identify by the inception model and variation of labels identified from the data points is abundant.

In WGAN-GP, WGAN-SN and WGAN-TS*, we observed that training suddenly deteriorated in some trials. We thus used early stopping on the basis of EWD, and the results of these methods shown in Table 4 are with early stopping.

The proposed method marked the worst in FID and the best in IS among all the methods compared. Certainly, the fake-data generated by the proposed method are non-sharp and do not resemble real-data points, but it seems that it is easy to distinguish them and they have diversity as digit images. If one wishes to produce higher FID results using the proposed method, transport cost should be considered in the desired space corresponding to FID.

