# OpenReview forum: "Generative model based on minimizing exact empirical Wasserstein distance"
_ICLR.cc/2019/Conference_

### Official Review · AnonReviewer2 · 2018-10-30
**Title claim seems wrong**

**Rating:** 3
**Confidence:** 4

**Review:**

The paper ‘Generative model based on minimizing exact empirical Wasserstein distance' proposes
a variant of Wasserstein GAN based on a primal version of the Wasserstein loss rather than the relying
on the classical Kantorovich-Rubinstein duality as first proposed by Arjovsky in the GAN context.
Comparisons with other variants of Wasserstein GAN is proposed on MNIST.

I see little novelty in the paper. The derivation of the primal version of the problem is already
given in
Cuturi, M., & Doucet, A. (2014, January). Fast computation of Wasserstein barycenters. In ICML (pp. 685-693).

Using optimal transport computed on batches rather the on the whole dataset is already used in (among
others)
 Genevay, A., Peyré, G., & Cuturi, M. (2017). Learning generative models with sinkhorn divergences. AISTATS
 Damodaran, B. B., Kellenberger, B., Flamary, R., Tuia, D., & Courty, N. (2018). DeepJDOT: Deep Joint distribution optimal transport for unsupervised domain adaptation. ECCV

Also, the claim that the exact empirical Wasserstein distance is optimized is not true. The gradients, evaluated on
batches, are biased. Unfortunately, the Wasserstein distance does not enjoy similar U-statistics as MMD. It is very
well described in the paper (Section 3):
https://openreview.net/pdf?id=S1m6h21Cb

Computing the gradients of Wasserstein on batches might be seen a kind of regularization, but it remains to be
proved and discussed.

Finally, the experimental validation appears insufficient to me (as only MNIST or toy datasets are considered).


Typos:
 Eq (1) and (2): when taken over the set of all Lipschitz-1 functions, the max should be a sup

---

### Official Review · AnonReviewer1 · 2018-11-02
**Review for "Generative model based on minimizing exact empirical Wasserstein distance".**

**Rating:** 2
**Confidence:** 5

**Review:**

The authors propose to estimate and minimize the empirical Wasserstein distance between batches of samples of real and fake data, then calculate a (sub) gradient of it with respect to the generator's parameters and use it to train generative models.

This is an approach that has been tried[1,2] (even with the addition of entropy regularization) and studied [1-5] extensively. It doesn't scale, and for extremely well understood reasons[2,3]. The bias of the empirical Wasserstein estimate requires an exponential number of samples as the number of dimensions increases to reach a certain amount of error [2-6]. Indeed, it requires an exponential number of samples to even differentiate between two batches of the same Gaussian[4]. On top of these arguments, the results do not suggest any new finding or that these theoretical limitations would not be relevant in practice. If the authors have results and design choices making this method work in a high dimensional problem such as LSUN, I will revise my review.

[1]: https://arxiv.org/abs/1706.00292
[2]: https://arxiv.org/abs/1708.02511
[3]: https://arxiv.org/abs/1712.07822
[4]: https://arxiv.org/abs/1703.00573
[5]: http://www.gatsby.ucl.ac.uk/~gretton/papers/SriFukGreSchetal12.pdf
[6]: https://www.sciencedirect.com/science/article/pii/0377042794900337

---

> ### Public Comment · ~Alexander_Mathiasen2 · 2020-05-12
> **Exponential Number of Samples to Differentiate Batches of same Gaussian.**
>
> > Indeed, it requires an exponential number of samples to even differentiate between two batches of the same Gaussian [4].
> Are you referring to Lemma 1?

---

### Official Review · AnonReviewer3 · 2018-11-02
**promising results and idea**

**Rating:** 5
**Confidence:** 2

**Review:**

The paper proposed to use the exact empirical Wasserstein distance to supervise the training of generative model. To this end, the authors formulated the optimal transport cost as a linear programming problem. The quantitative results-- empirical Wasserstein distance show the superiority of the proposed methods.

My concerns come from both theoretical and experimental aspects:
The linear-programming problem Eq.(4)-Eq.(7) has been studied in existing literature.
The contribution is about combining this existing method to supervise a standard neural network parametrized generator, so I am not quite sure if this contribution is sufficient for the ICLR submission.
In such a case, further experimental or theoretical study about the convergence of Algorithm 1 seems important to me.

As to the experiments, firstly, EWD seems to be a little bit biased since EWD is literally used to supervise the training of the proposed method.
Other quantitative metric studies can help justifying the improvement.
Also, given that the paper brings the WGAN family into comparison, the large scale image dataset should be included since WGAN have already demonstrated their success.

Last things, missing parentheses in step 8 of Algorithm 1 and overlength of url in references.

---

### Meta-Review · Area_Chair1 · 2018-12-05
**lack of novelty, variance in high dimensions**

**Confidence:** 5
**Recommendation:** Reject

**Metareview:**

This method proposes a primal approach to minimizing Wasserstein distance for generative models. It estimates WD by computing the exact WD between empirical distributions.

As the reviewers point out, the primal approach has been studied by other papers (which this submission doesn't cite, even in the revision), and suffers from a well-known problem of high variance. The authors have not responded to key criticisms of the reviewers. I don't think this work is ready for publication in ICLR.